# Biosurfactant Produced by *Bacillus subtilis* UCP 1533 Isolated from the Brazilian Semiarid Region: Characterization and Antimicrobial Potential

**DOI:** 10.3390/microorganisms13071548

**Published:** 2025-07-01

**Authors:** Antônio P. da C. Albuquerque, Hozana de S. Ferreira, Yali A. da Silva, Renata R. da Silva, Carlos V. A. de Lima, Leonie A. Sarubbo, Juliana M. Luna

**Affiliations:** 1School of Health and Life Sciences, Catholic University of Pernambuco (UNICAP), Príncipe Street, n. 526, Boa Vista, Recife 50050-900, Brazil; antonio.00000850826@unicap.br (A.P.d.C.A.); carlos.00000800050@unicap.br (C.V.A.d.L.); 2Northeast Biotechnology Network (RENORBIO), Federal Rural University of Pernambuco, Dom Manuel de Medeiros Street, Dois Irmãos, Recife 52171-900, Brazil; ferreira.hozanasouza@gmail.com (H.d.S.F.); yalialves@gmail.com (Y.A.d.S.); renatabiology2015@gmail.com (R.R.d.S.); 3School of Technology and Communication, Catholic University of Pernambuco (UNICAP), Príncipe Street, n. 526, Boa Vista, Recife 50050-900, Brazil; leonie.sarubbo@unicap.br

**Keywords:** antibacterial biosurfactant, *Bacillus subtilis*, antimicrobial resistance, caatinga soil

## Abstract

The increasing resistance of pathogenic microorganisms to antimicrobials has driven the search for safe and sustainable alternatives. In this context, microbial biosurfactants have gained prominence due to their antimicrobial activity, low toxicity, and high stability under extreme conditions. This study presents the production and characterization of a biosurfactant with antimicrobial potential, obtained from *Bacillus subtilis* isolated from soil, for application in the control of resistant strains. Bacterial identification was performed using mass spectrometry (MALDI-TOF), confirming it as *Bacillus subtilis*. The strain *B. subtilis* UCP 1533 was cultivated using different carbon sources (glucose, soybean oil, residual frying oil, and molasses) and nitrogen sources (ammonium chloride, sodium nitrate, urea, and peptone), with evaluations at 72, 96, and 120 h. The best condition involved a mineral medium supplemented with 2% soybean oil and 0.12% corn steep liquor, resulting in the production of 16 g·L^−1^ of biosurfactant, with a critical micelle concentration (CMC) of 0.3 g·L^−1^ and a reduction in water surface tension to 25 mN·m^−1^. The biosurfactant showed an emulsification index of 100% for used motor oil and ranged from 50% to 100% for different vegetable oils, maintaining stability across a wide range of pH, salinity, and temperature. FT-IR and NMR analyses confirmed its lipopeptide nature and anionic charge. Toxicity tests with *Tenebrio molitor* larvae showed 100% survival at all the tested concentrations. In phytotoxicity assays, seed germination rates above 90% were recorded for *Solanum lycopersicum* and *Lactuca sativa*. Antimicrobial tests revealed inhibitory activity against resistant strains of *Escherichia coli* and *Pseudomonas aeruginosa*, as well as against species of the genus *Candida* (*C. glabrata*, *C. lipolytica*, *C. bombicola*, and *C. guilliermondii*), highlighting the biosurfactant as a promising alternative in combating antimicrobial resistance (AMR). These results indicate the potential application of this biosurfactant in the development of antimicrobial agents for pharmaceutical formulations and sustainable strategies for phytopathogen control in agriculture.

## 1. Introduction

Antimicrobial resistance (AMR) is currently recognized as one of the main threats to global public health, jeopardizing the effectiveness of antibiotics and compromising significant advances in modern medicine. The indiscriminate use of antimicrobial agents in clinical, veterinary, and agricultural settings has directly contributed to the emergence and spread of multidrug-resistant microorganisms (MDR), resulting in infections that are difficult to treat, increased hospital costs, and high morbidity and mortality rates [1,2]. Considering this alarming scenario, there is a growing need to develop new therapeutic strategies and alternative agents capable of overcoming the current mechanisms of bacterial resistance [3].

Antimicrobial resistance (AMR) occurs when microorganisms, such as bacteria and fungi, acquire the ability to withstand the effects of antimicrobial agents that were previously effective against them. This phenomenon is primarily driven by the indiscriminate and inappropriate use of these drugs, which exerts selective pressure, favoring the survival and proliferation of resistant strains through natural selection [4,5]. Preventing AMR, therefore, requires integrated and global strategies, focusing on the rational use of antibiotics in humans and animals, as well as hygiene and sanitation measures [6].

The “One Health” approach emphasizes the importance of reducing unnecessary antimicrobial use across all sectors. However, challenges such as inconsistent regulations and indiscriminate use in agriculture hinder the implementation of effective policy. Although the development of new antimicrobials is essential, it faces significant economic barriers, such as low financial returns due to restricted use, high research and development costs, and a lack of sustainable business models, which discourage investment by major pharmaceutical companies. To address this issue, investments are being made in alternative strategies, such as combined therapies, new molecular targets, bacteriophages, and natural compounds, to prolong treatment efficacy and contain the spread of resistance [4,7,8].

Several approaches have been employed to combat antimicrobial resistance, among which biosurfactants have shown promise as an alternative in fighting pathogenic and resistant microorganisms [9].

Biosurfactants are surface-active compounds of biological origin produced by various microorganisms, such as bacteria, yeasts, and filamentous fungi, synthesized at the end of the exponential growth phase of these microorganisms, and secreted extracellularly or remain adhered to cell surfaces [10,11]. They possess amphiphilic properties, meaning they simultaneously have hydrophilic and hydrophobic regions, which gives them the ability to reduce surface and interfacial tensions. Due to their biodegradability, low toxicity, and efficacy under extreme conditions of temperature, pH, and salinity, biosurfactants are considered sustainable alternatives to synthetic surfactants in various industrial applications, including pharmaceutical, cosmetic, food, and environmental remediation [4,12,13].

In addition to their surface-active and emulsifying properties, many biosurfactants exhibit proven antimicrobial activity by destabilizing the cell membrane, inhibiting bacterial adhesion, and dispersing biofilms [14,15]. Recent research has highlighted the effectiveness of these compounds against both Gram-positive and Gram-negative bacteria, including strains resistant to multiple classes of antibiotics [16]. However, large-scale production, standardization of production and development processes, and structural and functional characterization of these biosurfactants still represent significant challenges in the field [17].

The genus *Bacillus*, especially the species *Bacillus subtilis*, has been widely studied as a producer of biosurfactants, with an emphasis on lipopeptides such as surfactin, which possesses recognized antimicrobial activity and the ability to reduce the surface tension of the medium [18]. In addition to its Generally Recognized as Safe (GRAS) profile, *B. subtilis* exhibits rapid growth and genetic stability, characteristics that favor its industrial application [5]. Despite this, investigations into the efficacy of biosurfactants produced by different strains of *B. subtilis*, particularly against multidrug-resistant pathogens, are still scarce and deserve further attention.

In this context, this study aimed to isolate, identify, and produce a biosurfactant from *Bacillus subtilis* UCP 1533, a strain obtained from Catimbau National Park, located in the Brazilian Caatinga biome—an ecologically unique and underexplored semi-arid region. By investigating a microbial strain adapted to extreme environmental conditions, this work addresses a critical gap in the literature regarding the biosurfactant-producing potential of *B. subtilis* in non-conventional ecosystems. The biosurfactant was thoroughly characterized with respect to its physicochemical properties, including surface tension reduction and emulsifying capacity, antimicrobial activity against resistant pathogens, and toxicity in biological models. These findings highlight the potential of microbial resources from neglected biomes for the development of novel and sustainable bioactive compounds with industrial and biomedical relevance.

## 2. Materials and Methods

### 2.1. Soil Collection and Microorganism Identification

The microorganism used as the biosurfactant producer was isolated from a soil sample collected from Catimbau National Park, located in the municipality of Buíque, Pernambuco, Brazil. This region has sparse, low vegetation typical of the Caatinga biome. Soil was collected at approximately 10 cm depth using a previously sterilized gardening shovel. The samples were placed in sterile Falcon tubes and transported to the Bioengineering Laboratory of the Catholic University of Pernambuco, where they were processed. Under controlled laboratory conditions, 1 g of the soil sample was suspended in 9 mL of sterile saline solution (0.85% (*w*/*v*) NaCl) and homogenized using a vortex mixer for 1 min. Aliquots were plated onto Petri dishes containing Nutrient Agar medium and incubated at 30 °C for 24 to 48 h to isolate distinct bacterial colonies. Colonies with distinct morphologies were successively streaked until pure cultures were obtained, which were subsequently identified by mass spectrometry (MALDI-TOF MS) and confirmed through Gram staining and biochemical tests. The isolated strain was maintained in test tubes containing Nutrient Agar medium, refrigerated at 5 °C, with subculturing performed every 30 days to ensure cell viability.

#### 2.1.1. Microorganism Identification by MALDI-TOF MS

Bacterial identification was performed by mass spectrometry using the MALDI-TOF MS technique (MALDI Sirius, Bruker Daltonics Inc., Bremen, Germany), following the methodology described by Lima-Neto et al. [19]. The isolate was cultured on Sabouraud Dextrose Agar and incubated at 30 °C for 24 h. For the ionization of ribosomal and structural proteins, a small amount of biomass was applied in duplicate onto polished steel target spots (Polished Steel Target, Bruker Daltonics, Bremen, Germany). Then, 1 µL of 70% formic acid was added, and after drying, 1 µL of a matrix composed of α-cyano-4-hydroxycinnamic acid (CHCA) was applied. The matrix was prepared from a CHCA solution at 75 mg/mL in ethanol, water, and acetonitrile (1:1:1), containing 0.03% trifluoroacetic acid (TFA). Analyses were conducted in linear mode, with a scanning range between 2.000 and 20.000 Da, a pulse of 104 ns, and a voltage of +20 kV. Final spectra were obtained by summing 240 laser shots, that is, 40 shots per profile and six profiles per sample. The peak list was exported to the MALDI Biotyper HT module software (Bruker Daltonics, Bremen, Germany), where the final identification was performed based on the set of peaks present in the spectrum.

#### 2.1.2. Gram Staining

Was performed for morphological characterization and differentiation of bacterial cell walls. The bacterial culture smear was heat-fixed on a slide and subjected to sequential staining with crystal violet for 1 min, followed by the application of Lugol’s solution for 1 min. The slides were then washed with 95% ethyl alcohol for approximately 20 s and counterstained with safranin for 30 s. The samples were then rinsed with distilled water and observed under a light microscope at 1000× magnification (oil immersion). The evaluation was based on the retention of the dyes by the cell wall structures.

#### 2.1.3. Spore Staining by Wirtz-Conklin Technique

Prior to staining, the bacterial strain was subjected to heat stress at 44 °C to induce spore formation. Next, a thin, uniform bacterial smear was prepared and fixed on a slide. The sample was then covered with Malachite Green dye and subjected to indirect heating by placing the slide over the mouth of a beaker with heated water until vapor was released for approximately five minutes. During this period, the dye was reapplied at intervals of one to two minutes, totaling three to four applications, to keep the slide consistently moist and facilitate dye penetration into the spores. After heating, the slide was gently rinsed with running water to remove excess dye, avoiding thermal shocks that could compromise the integrity of the slide. Subsequently, the counterstain safranin was applied for 30 s, followed by gentle rinsing with water.

#### 2.1.4. Catalase Test

Was performed to verify the presence of catalase in the bacterial sample. For this, a small portion of the colony was transferred to a glass slide using a platinum loop. Two to three drops of 3% (*v*/*v*) hydrogen peroxide (H_2_O_2_) solution were added directly to the sample. Immediate observation of the reaction was performed to detect the release of bubbles, indicative of enzymatic activity, according to protocols described in standard microbiology manuals [20,21].

### 2.2. Preparation and Growth of the Inoculum

The *Bacillus subtilis* inoculum was transferred from a test tube containing Nutrient Agar to Erlenmeyer flasks containing Nutrient Broth medium, which was previously sterilized in an autoclave at 121 °C for 20 min. The flasks were incubated in an orbital shaker at 200 rpm and 35 °C for 24 h. After this period, the inoculum was standardized to an optical density of 0.7, corresponding to an inoculum concentration of 10^7^ CFU/mL, measured at 600 nm. Subsequently, the standardized inoculum was added to the production medium at a proportion of 2.0% (*v*/*v*).

### 2.3. Selection of Cultivation Conditions for Biosurfactant Production

The microorganism was cultured in a mineral medium containing: 0.06% MgSO_4_·7H_2_O, 0.087% K_2_HPO_4_, 0.02% KCl, 0.01% NaCl, 0.65% Tris(hydroxymethyl)aminomethane, and 0.005% yeast extract. Fermentation was carried out in 250 mL Erlenmeyer flasks containing 100 mL of the medium, with the pH previously adjusted to 7.0 ± 0.2, under orbital shaking at 200 rpm at 35 °C (Model TECNAL TE-424).

The optimization of the culture medium was performed through an integrated analysis of the effects of carbon and nitrogen sources on fermentation time, aiming to maximize biosurfactant production. Different carbon sources (glucose, molasses, soybean oil, and residual frying oil at 2%) and subsequently nitrogen sources (corn steep liquor, urea, ammonium chloride, sodium nitrate, and peptone at 0.12%) were tested, with cultures conducted for 72, 96, and 120 h. The surface tension of the medium was used as a criterion to select the best production conditions. (Figure 1).

### 2.4. Biosurfactant Production

Fermentations for biosurfactant production were carried out in 2 L Erlenmeyer flasks containing 1 L of mineral medium supplemented with 2.0% (*v*/*v*) carbon source and 0.12% (*v*/*v*) nitrogen source, selected based on the results obtained during the evaluation of the different sources tested. Subsequently, the medium was sterilized in an autoclave at 121 °C for 20 min and, after cooling, inoculated with a previously standardized cell suspension (10^7^ CFU/mL). Fermentations were conducted under orbital shaking at 200 rpm and 35 °C for the optimal time determined during the cultivation selection assays.

### 2.5. Determination of Surface Tension and Critical Micelle Concentration (CMC)

The surface tension of the biosurfactant was determined in the cell-free metabolic liquid obtained after centrifugation. The measurements were performed at room temperature using the du Noüy ring method with a Sigma 700 tensiometer (KSV Instruments Ltd., Helsinki, Finland). To determine the Critical Micelle Concentration (CMC), 0.1 g of the purified biosurfactant was dissolved in an initial solution with a concentration of 5 g·L^−1^. From this solution, successive dilutions were prepared in distilled water, and the surface tension was measured again using the du Noüy ring method.

### 2.6. Determination of Emulsifying Activity

The emulsifying activity of the biosurfactant was determined using the method described by Cooper and Goldenberg [22]. In this method, 2 mL of hydrocarbon (motor oil, soybean oil, and sunflower oil) was added to 2 mL of cell-free metabolic liquid in a test tube, and the mixture was vortexed (K45-2820, KASVI, São José dos Pinhais, Brazil) for 2 min. The emulsions were then left to stand for 24 h at 27 °C, and the Emulsification Index (EI24) was calculated as a percentage using the following formula:(1)$${\mathrm{IE}}_{24}\;(\%)=\frac{he}{ht}\times 100\;$$
where *he* is the height of emulsification and *ht* is the total height of the mixture, both expressed in cm.

### 2.7. Evaluation of Biosurfactant Stability (Effects of pH, Temperature, and NaCl)

The stability of the produced biosurfactant was evaluated under three environmental conditions: temperature, salinity, and pH. To this end, the effects of different temperatures (5 °C, 50 °C, 70 °C, and 100 °C), NaCl concentrations (2.0%, 4.0%, 6.0%, 8.0%, 10.0%, and 12.0% (*w*/*v*)), and pH ranges (2.0, 4.0, 6.0, 8.0, 10.0, and 12.0) were analyzed separately. The biosurfactant was then subjected to surface tension and emulsifying activity analyses to assess its stability. All assays were performed in triplicate.

### 2.8. Biosurfactant Isolation

The extraction of the biosurfactant was performed using a volumetric ratio of 1:4 (*v*/*v*) of crude biosurfactant to solvent (ethyl acetate). The process was repeated twice, followed by centrifugation at 4500 rpm for 15 min to separate the two phases. The organic fraction was transferred to a separatory funnel, washed with saturated sodium chloride (NaCl) solution, and the resulting aqueous phase was discarded. The organic solvent was then dried with anhydrous sodium sulfate (Na_2_SO_4_), filtered, and evaporated using a heated plate to obtain the isolated biosurfactant.

### 2.9. Physicochemical Characterization of the Biosurfactant

For the chemical composition characterization of the biosurfactant, Nuclear Magnetic Resonance (NMR) Spectroscopy and Fourier Transform Infrared (FT-IR) Spectroscopy techniques were employed. The purified sample was dissolved in deuterated chloroform (CDCl_3_) and analyzed using an Agilent spectrometer (Santa Clara, CA, USA) operating at 300.13 MHz for the ^1^H nucleus and 75.47 MHz for ^13^C. Spectra were obtained on the chemical shift scale (δ), expressed in parts per million (ppm), using tetramethylsilane (TMS) as the internal standard.

Infrared spectroscopy analysis was performed using an FT-IR spectrometer (Spectrum 400, Perkin Elmer, Shelton, CT, USA) in the spectral range of 4000 to 400 cm^−1^. The accuracy was maintained within the wavelength range of −0.1 to +0.1 cm^−1^.

### 2.10. Ionic Charge Determination

The ionic charge of the biosurfactant was determined using the agar double-diffusion technique [23]. The test tubes were filled with the biosurfactant solution and compounds of known ionic charges. Sodium Dodecyl Sulfate (SDS) at 20 mM was used as an anionic reference, and Barium Chloride at 50 mM as a cationic reference. The formation of precipitation lines between the tubes indicates electrostatic interactions and, therefore, the ionic nature of the biosurfactant. The assay was monitored for 48 h at room temperature.

### 2.11. Toxicological Analysis

#### 2.11.1. Toxicity Assessment of the Biosurfactant in *Tenebrio molitor*

*Tenebrio molitor* larvae were randomly distributed in Petri dishes in groups of five individuals. A volume of 10 µL of the biosurfactant was injected into the ventral membrane between the second and third abdominal segments (tail-to-head direction) at concentrations corresponding to ½ CMC (0.15 g·L^−1^), CMC (0.3 g·L^−1^), and 2× CMC (0.6 g·L^−1^). Larval viability was monitored at intervals of 24, 48, 72, 96, and 120 h, with the absence of response to mechanical stimulus considered indicative of death. Larvae inoculated with PBS were used as negative controls. The obtained data were represented by survival curves over time [24].

#### 2.11.2. Phytotoxicity in *Solanum lycopersicon* and *Lactuca sativa*

The phytotoxicity of the biosurfactant was analyzed by exposing seeds of *Solanum lycopersicum* (tomato) and *Lactuca sativa* (lettuce) to different concentrations of the biosurfactant. Test solutions were prepared with concentrations of ½ CMC (0.15 g·L^−1^), CMC (0.3 g·L^−1^), and 2× CMC (0.6 g·L^−1^) in distilled water. The seeds were incubated for five days in the dark, and the effects on germination and root growth were evaluated according to the methodology described by Tiquia, Tam, and Hodgkiss [25].

### 2.12. Disk Diffusion Assay of the Biosurfactant

The antimicrobial activity of the biosurfactant was evaluated using the Kirby-Bauer disk diffusion method. The assay was performed against two Gram-negative bacteria: *Escherichia coli* and *Pseudomonas aeruginosa* (bacterial strains isolated from clinical samples from a reference hospital), cultivated on Mueller-Hinton agar medium (Sigma-Aldrich), and four yeasts: *Candida glabrata*, *Candida lipolytica*, *Candida bombicola*, and *Candida guilliermondii*, were cultivated on Sabouraud Dextrose agar medium. The pre-inoculum of the microorganisms was adjusted to a turbidity equivalent to 0.5 McFarland scale, corresponding to approximately 1.5 × 10^8^ CFU/mL. Sterile filter paper disks (Whatman No. 1, 5 mm diameter) were impregnated with 20 µL of the biosurfactant and placed on the surface of Petri plates previously inoculated with the target microorganisms. Mueller-Hinton agar was used for bacteria, and Sabouraud agar for yeast. The plates with bacteria were incubated at 37 °C for 24 h, while the plates with yeast were incubated at 28 °C for the same period. After incubation, the inhibition halos were measured, and the results were expressed as the percentage of inhibition, as described by Leyton et al. [26].

### 2.13. Statistical Analysis

The data obtained were statistically analyzed using Statistica software (version 7.0). Analysis of variance (ANOVA) was used to verify significant differences between groups. All experiments were conducted in triplicate, and the results are presented as mean ± standard deviation (n = 3). A 95% confidence interval was considered, with a significance level of 5% (*p* < 0.05).

## 3. Results and Discussion

### 3.1. Microorganism Identification

Conventional bacterial identification at the species level is a time-consuming and complex process based on phenotypic characteristics observed during cultivation, such as staining and growth rates. Additionally, it involves biochemical tests and the use of specific substrates, which often do not provide precise identification [27]. In this context, MALDI-TOF mass spectrometry has proven to be effective in identifying various bacterial species. This technique generates a mass spectrum that represents the proteomic profile of the analyzed microorganism, acting like a “fingerprint.” These profiles are characteristic of each species, allowing for the differentiation of specific peaks associated with certain genera and species [28].

The identification of the bacterial isolate was carried out by comparing the obtained spectra with the Bruker Biotyper database. The identity score was 1.87. According to the system criteria, scores between 1.700 and 1.999 indicate probable identification at the genus level and are considered reliable for this taxonomic rank [29]. Therefore, the sample was assigned to the genus *Bacillus*, with *Bacillus subtilis* as the most likely species. The bacterial strain was cataloged as *Bacillus subtilis* UCP 1533 by the Multiuser Center for Biomolecule Analysis and Surface Characterization of Materials at the Catholic University of Pernambuco.

#### 3.1.1. Gram Staining

Gram staining is a widely used method for classifying bacteria into two major groups: Gram-positive and Gram-negative. This test is essential for confirming bacterial isolates.

Gram staining revealed Gram-positive rod-shaped cells, consistent with the characteristic morphology of the species *Bacillus subtilis* (Figure 2).

Determining the Gram reaction of *Bacillus subtilis* is crucial because it reflects the fundamental structural and physiological characteristics of the species. As a Gram-positive bacterium, *B. subtilis* possesses a thick peptidoglycan cell wall that contributes to its resistance to environmental stresses, including desiccation and chemical agents. This structural trait is also related to its ability to form endospores, an essential survival strategy under harsh conditions. This aids in the differentiation of Gram-negative microorganisms commonly found in soil environments.

#### 3.1.2. Spore Staining by the Wirtz-Conklin Technique

Wirtz-Conklin staining allowed the visualization of typical spore structures in the analyzed sample, confirming the microorganism’s ability to form endospores (Figure 3).

The genus *Bacillus* is recognized for its ability to form endospores, which are highly specialized structures that provide resistance to adverse environmental conditions, such as nutrient scarcity and thermal stress [30]. Spore formation is characterized by complex cellular reorganization and the accumulation of compounds such as calcium and dipicolinic acid, which promote cell dehydration and enhance heat resistance [31,32].

*Bacillus subtilis* is a widely used model species in sporulation studies due to its well-documented ability to produce robust and viable spores [33,34]. Each vegetative cell can produce a single spore that is released after cell lysis. The presence of these structures, evidenced by differential staining, is associated with an adaptation and survival mechanism against environmental stress and is a highly genetically regulated process. The findings of this study corroborate the current literature, reinforcing the use of *B. subtilis* as a model microorganism in research on bacterial resistance and spore detection techniques [33,35].

#### 3.1.3. Catalase Test

The catalase test aims to verify the presence of catalase in the sample. The formation of bubbles was considered a positive result, while the absence of bubbles was considered a negative result (Figure 4).

The catalase activity observed in the isolate was consistent with that reported in recent studies on *Bacillus subtilis*. This species is known to produce catalases, such as KatA, which play a crucial role in protecting against oxidative stress [36]. Research indicates that *B. subtilis* can increase catalase production in response to oxidative stress, as demonstrated by Bano et al. [37], who observed higher catalase activity under stressed conditions.

Therefore, the positive catalase test, together with Gram staining, the Wirtz-Conklin technique, and data obtained by mass spectrometry (MALDI-TOF MS), provides supportive evidence for the identification of the isolate as *Bacillus subtilis*.

### 3.2. Selection of Cultivation Conditions for Biosurfactant Production

The use of alternative and low-cost sources, such as industrial waste, significantly contributes to the economic viability of fermentation processes, making biosurfactant production more sustainable and competitive. Moreover, the choice of carbon and nitrogen sources can directly influence the physicochemical properties of the obtained biosurfactants, allowing modulation of their characteristics and, consequently, optimization of their applications across different industrial sectors [38,39].

In this context, *B. subtilis* UCP 1533 was cultivated in mineral media supplemented with combinations of substrates as carbon and nitrogen sources to provide energy and structural elements for microbial growth and biosynthesis of biomolecules.

Based on the results obtained, soybean oil (40.5 mN·m^−1^) and corn steep liquor (34.3 mN·m^−1^) were selected as the carbon and nitrogen sources, respectively, for the culture medium. The results also indicated that extending the cultivation time to 120 h significantly reduced the surface tension.

Although *B. subtilis* UCP 1533 demonstrated the ability to grow on all tested sources (Figure 5 and Figure 6), the selection of the most suitable sources and the optimal cultivation time was based on its performance in reducing the surface tension of the medium (Table 1). This parameter is considered a key indicator of biosurfactant efficiency, as it directly reflects the ability of the compound to reduce intermolecular forces at liquid interfaces, thereby enhancing the emulsification, solubilization, and dispersion of hydrophobic compounds [40].

From this perspective, one of the advantages of soybean oil and other vegetable oils is their higher carbon content compared to sugars, which leads to a greater yield of the compound to be produced [41]. Corn steep liquor, on the other hand, is predominantly composed of various amino acids and reduced sugars. Analyses carried out on this byproduct, regarding the contents of free and total amino acids, total sugars, reducing sugars, total sulfites, and total nitrogen, demonstrate its importance as an efficient substrate for fermentation processes [42].

Selva Filho et al. [43] maximized the production of a biosurfactant produced by the yeast *Starmerella bombicola* ATCC 222214 through the investigation of different carbon and nitrogen sources, as well as various cultivation conditions. The selected mineral medium for production was supplemented with 2.0% potato peel flour, 5.0% residual canola frying oil, and 0.20% urea after 180 h of cultivation.

In recent studies, Ciurko et al. [44] reported a carefully monitored fermentation process to characterize the efficiency and functionality of the biosurfactant produced by *Bacillus subtilis* #309. Maximum surfactin production was achieved in media supplemented with sunflower and rapeseed cakes. In parallel, a progressive reduction in the surface tension was observed.

According to the literature, *Bacillus licheniformis* EL3 was investigated for biosurfactant synthesis in a bioreactor using a mineral medium enriched with glucose as the carbon source and NaNO_3_ and NH_4_Cl as nitrogen sources. After optimizing the operational parameters, including glucose concentration, *B. licheniformis* EL3 produced biosurfactants within 43 h of cultivation. The purified compound exhibited excellent surface activity characteristics, with a significant reduction in surface tension to a minimum value of 29 mN·m^−1^ [17].

### 3.3. Biosurfactant Production and Yield

The microbial surfactant developed in this study was formulated using a mineral medium supplemented with 2% soybean oil and 0.12% corn steep liquor. After 120 h of cultivation at 200 rpm and 35 °C, the cell-free metabolic liquid produced by Bacillus subtilis UCP 1533 reduced the surface tension of water from 72 to 31.8 mN·m^−1^. Following the isolation process, a yield of approximately 16 g·L^−1^ was obtained.

Several factors can influence the synthesis, quality, and quantity of the surfactant biomolecule produced. These factors may vary depending on the specific microorganism, the composition of the culture medium, and the type of biosurfactant formulated. In addition to carbon and nitrogen sources significantly affecting production, they also impact the yield. Other parameters that can influence microbial metabolism and biosurfactant synthesis include physicochemical factors that define the operational conditions, such as growth pH, temperature, aeration, and oxygen availability [45]. According to Cooper and Goldenberg [22], a microorganism is considered promising for biosurfactant production when it can reduce the surface tension to 40 mN·m^−1^ or lower.

According to the literature, other *B. subtilis* strains are also biosurfactant producers. Wu et al. [46] investigated the SL strain, identified as *Bacillus subtilis* through molecular methods, which was able to produce 1.32 g·L^−1^ of biosurfactant using sucrose as the sole carbon source after 72 h of fermentation. The surface tension of the cell-free metabolic liquid was 25.6 mN·m^−1^.

Hu et al. [47] carried out large-scale production of a biosurfactant using *Bacillus subtilis* ATCC 21332 and tuna fish residue as substrate, and as a result, they achieved the highest surfactin productivity at pilot scale, with a yield of 0.274 g·L^−1^. Liu et al. [48] investigated the influence of hydrocarbon-based carbon sources on biosurfactant production by *Bacillus licheniformis* L20 and obtained the highest concentration in LB/MSS-K medium, which was 1.225 g·L^−1^.

In this context, considering the high yield of biosurfactant obtained in the production process of this study, it is noteworthy that the results are approximately 12 times higher than that reported by Wu et al. [46] and almost 60 times greater than that achieved by Hu et al. [47], therefore, superior to those reported in the literature. This indicates that the operational conditions and formulation of the medium used are considerably efficient, favoring both the synthesis and productivity of the biosurfactant.

### 3.4. Physicochemical Properties of the Biosurfactant

The physicochemical properties of the biosurfactant were evaluated to determine its biotechnological potential. The emulsification capacity and critical micelle concentration were analyzed in the presence of a hydrophobic substrate, providing essential information regarding its efficiency and applicability.

The emulsification capacity is frequently used as an appropriate technique for detecting surfactant compounds. Moreover, this property has also been employed as a measurement tool to determine the ability of the biosurfactant to form and stabilize emulsions of different hydrophobic substrates, considering its potential practical applications in various fields. In addition to reducing surface and interfacial tensions, biosurfactants generally exhibit emulsifying capacity [49,50].

In the present study, the biosurfactant produced by *B. subtilis* UCP 1533 showed an emulsification index of 100% in the presence of burned motor oil, demonstrating its high efficiency in the formation and stabilization of emulsions. Similar results were observed by Singh et al. [51], who reported an emulsification index of 85.63% for a biosurfactant produced by *Bacillus megaterium*. In comparison, another study conducted by Maia et al. [52] described an emulsification of 95% using burned motor oil and a biosurfactant produced by *Bacillus subtilis*.

The critical micelle concentration (CMC) is a fundamental parameter for characterizing the surface-active activity and solubility of biosurfactants in aqueous media. The CMC is determined by the molecular structure and chemical composition of the biosurfactant and is also influenced by environmental conditions such as temperature, ionic strength, and the presence of organic additives [53].

The behavior of surface tension as a function of biosurfactant concentration in the medium is illustrated in Figure 7. As indicated in Figure 7, a concentration of 0.3 g·L^−1^ reduced the surface tension of water from 72 to 25.704 ± 0.28 mN·m^−1^, which corresponds to the CMC for the biosurfactant produced by *Bacillus subtilis* UCP 1533 in medium supplemented with corn steep liquor and soybean oil. This result was superior to that obtained by Ja’afaru et al. [54], who reported a CMC of 6.5 g·L^−1^ for a biosurfactant produced by *Bacillus subtilis*, with a reduction of water surface tension to 29.30 mN·m^−1^. Similarly to the results obtained in this study, Mnif et al. [55] reported a CMC of 0.35 g·L^−1^ and a reduction in water surface tension to 32 mN·m^−1^ using a biosurfactant produced by *Bacillus subtilis*.

### 3.5. Biosurfactant Stability

According to the literature, long-term stability is a fundamental requirement for the development and commercialization of new biotechnological products, whose functionality should not be significantly altered by sensitivity to variations in pH, temperature, and salinity typical of industrial environments [56].

The results presented in Table 2 show that the biosurfactant produced by *B. subtilis* UCP 1533 exhibited moderate stability under various tested conditions. Regarding temperature, the surface tension remained low throughout the evaluated range (5–100 °C), with the lowest values observed at 50 °C (26.0 mN·m^−1^) and 100 °C (26.2 mN·m^−1^). This thermal stability demonstrates its suitability for heating processes. In the pH assays, the biosurfactant showed sensitivity to the tested conditions (2–12), suggesting that its stability is influenced by pH variations. Under the evaluated saline conditions, the biosurfactant exhibited excellent stability, maintaining a surface tension between 27.4 and 27.9 mN·m^−1^ even at high NaCl concentrations (up to 12%).

Emulsification is characterized by the dispersion of one liquid phase into another, resulting in the formation of numerous small droplets. Efficient emulsifiers are characterized by a high capacity to reduce interfacial tension, rapid adsorption kinetics, and a high degree of coverage at water-oil interfaces after adsorption [57]. In this context, the emulsification stability of motor oil, soybean oil, and sunflower oil was evaluated using the biosurfactant produced by *B. subtilis* UCP 1533. The assays were conducted under the same conditions used to assess surface tension stability, regarding the influence of wide ranges of pH, temperature, and salinity.

Table 3 presents the results obtained for the emulsification stability of the biosurfactant produced by *B. subtilis* UCP 1533 at different pH values. According to the data, the biosurfactant exhibited high emulsifying stability across the tested pH range (2–12), highlighting its potential for application in environments with varying chemical conditions. The emulsification capacity of motor oil was 100% under all tested conditions, suggesting a strong hydrophobic affinity and efficiency in stabilizing emulsions involving nonpolar compounds, whether in acidic or alkaline media. Although the emulsification efficiencies of soybean and sunflower oils were slightly lower, the emulsions remained stable, with efficiencies varying from 50 to 55% and 35 to 40% for soybean and sunflower oils, respectively.

The results presented in Table 4 demonstrate that the biosurfactant produced by *B. subtilis* UCP 1533 exhibited high emulsification stability across a wide temperature range of 5 °C to 100 °C. The emulsification efficiency of motor oil remained at 100% under all tested conditions, indicating that the structure and functionality of the biosurfactant were not affected by thermal variations, even at high temperatures. This behavior highlights the remarkable thermal stability, which is essential for industrial applications operating under extreme heat or cold conditions. Soybean and sunflower oils showed moderate variations in emulsification efficiency with temperature. Soybean oil emulsification ranged between 55 and 60%, while sunflower oil emulsification increased from 35 to 55% with increasing temperature.

The results in Table 5 indicate that the biosurfactant produced by *B. subtilis* UCP 1533 exhibited high emulsifying stability across different salt (NaCl) concentrations, ranging from 2 to 12%. The emulsification of motor oil remained constant (100%) at all evaluated concentrations, demonstrating that the presence of salt does not compromise the ability of the biosurfactant to stabilize hydrophobic emulsions, highlighting its excellent salt resistance. Soybean oil exhibited relatively stable emulsification, varying between 40 and 45%, with a slight increase at higher salt concentrations. Sunflower oil remained practically constant between 30 and 35% across all tested NaCl concentrations.

The literature data corroborate the stability results observed for the biosurfactant from *B. subtilis* UCP 1533. Jumpathong et al. [58] isolated and characterized *Bacillus velezensis* PW192, a producer of antifungal biosurfactants. The biosurfactant exhibited an emulsifying capacity of up to 60% and reduced the surface tension of distilled water from 72 mN·m^−1^ to 21 mN·m^−1^. The surfactant properties remained stable across a wide pH range (6 to 10), at high temperatures (up to 100 °C), and under high salinity conditions, with NaCl concentrations up to 12%. Wu et al. [46] showed that *Bacillus subtilis* SL presented excellent emulsification with crude oil; furthermore, the biosurfactant exhibited excellent surface activity across broad pH ranges (5–12), NaCl concentrations (10%), and at elevated temperatures (120 °C). The glycolipid biosurfactant obtained from the endophytic bacterial strain *Bacillus pumilus* 2A, produced by Marchut-Mikołajczyk et al. [13], demonstrated high and long-term thermostability, a surface tension of 47.7 mN·m^−1^, and an emulsion index of 69.11%.

### 3.6. Determination of Ionic Charge

The agar double diffusion assay demonstrated the formation of precipitation lines between the biosurfactant produced by *B. subtilis* UCP 1533 and the cationic compound barium chloride, while no precipitation was observed in the interaction with sodium dodecyl sulfate, an anionic compound. Thus, under the experimental conditions of this study, the assay indicated that the biosurfactant had an anionic character. Literature reports that Other biosurfactants also possess an anionic charge, as is the case with the biosurfactant from *B. subtilis* UCP 0999 [59].

### 3.7. Structural Characterization of the Biosurfactant Produced by Bacillus subtilis UCP 1533

The biosurfactant produced by submerged fermentation with *Bacillus subtilis* UCP 1533, using corn steep liquor and soybean oil as combined sources of carbon and nitrogen, was structurally characterized using Fourier-transform infrared spectroscopy (FT-IR) and proton and carbon nuclear magnetic resonance (^1^H and ^13^C NMR).

The FT-IR spectrum (Figure 8) shows a broad band at 3398 cm^−1^, characteristic of the stretching vibration of hydroxyl (–OH) and amide (–NH) groups, which are common in peptide compounds. The peaks at 2924 cm^−1^ and 2853 cm^−1^ are attributed to the stretching of aliphatic C–H bonds, suggesting the hydrophobic chains of fatty acids, which are part of the lipid portion of the molecule.

The well-defined peak at 1743 cm^−1^ corresponds to the stretching vibration of the carbonyl groups (C=O) of esters, while the band at 1629 cm^−1^ indicates amide bonds (C=O and N–H) typical of peptides. The presence of bands at 1460 cm^−1^ and 1377 cm^−1^ further supports the presence of CH_3_ and CH_2_ groups, again associated with fatty chains.

The bands from 1161 to 1064 cm^−1^ can be attributed to the stretching vibrations of C–O and C–N bonds, compatible with the presence of esters and peptide bonds, while the band at 720 cm^−1^ is related to the out-of-plane bending of long aliphatic chains.

Similarly, FT-IR analysis conducted by Nurhasanah et al. [60] on a biosurfactant from *Bacillus* sp. ALPD1 indicated the presence of peptide groups, with stretching bands of N–H, C=O, and C–O–N. Aliphatic C–H stretching vibrations were also observed, with intensifications in other regions of the spectrum, suggesting the presence of aliphatic chains typical of lipopeptide compounds.

Umar et al. [61], when analyzing by FT-IR a biosurfactant obtained from *Bacillus subtilis* SNW3, identified the presence of aliphatic amine groups (C–N) and ester-type carbonyl groups, associated with the peptide portion, as well as bands attributed to the stretching of aliphatic C–H bonds (CH_2_–CH_3_). Hydroxyl groups (–OH) and possible phenols were also observed. These results were compared with the spectrum of standard surfactin (Sigma), which confirmed the lipopeptide nature of the analyzed biosurfactant.

The ^1^H NMR spectrum (400 MHz) (Figure 9) revealed intense signals in the range of 0.8–1.5 ppm, corresponding to the hydrogens of the methyl and methylene groups from the aliphatic chains, which are typical of saturated fatty acids. The presence of signals between 2.0 and 2.5 ppm suggests methylenes adjacent to carbonyls (–CH_2_–CO–), as occurs in amino acids and peptides. These data are consistent with the presence of peptide bonds in the samples. The region from 3.0–4.5 ppm showed multiple signals of hydrogens attached to oxygenated or nitrogenated carbons (C–O, C–N), which are common in modified amino acid structures. A small signal at 5.3 ppm indicates possible vinylic hydrogens (–CH=CH–), suggesting unsaturation in the fatty acid chains.

Similarly, Durval et al. [62] conducted the structural characterization of a lipopeptide produced by *Bacillus cereus* UCP 1615 cultured in a medium containing mineral salts, peptone, and residual soybean oil. Through ^1^H NMR analysis, the authors identified signals corresponding to methyl groups (0–1.0 ppm), aliphatic carbons (1.2–1.4 ppm), and hydrogens near double bonds or carbonyls (1.9–2.1 ppm). Similarly, Kumari et al. [63], when characterizing a lipopeptide produced by *Brevibacterium casei* LS14 cultured in a medium with residual glycerol, peptone, and NaCl, observed signals compatible with long aliphatic chains, as well as peaks indicative of carbonyl groups adjacent to methylenes, which are typical features of the lipid portion of lipopeptides.

In the ^13^C NMR spectrum (100 MHz) (Figure 10), signals were observed between 10 and 40 ppm, which are typical of aliphatic carbons. A range of 60–80 ppm may indicate carbons bonded to oxygen or nitrogen, as found in hydroxylated amino acids. The signals between 170 and 180 ppm are clearly attributable to carbonyl (C=O) groups of fatty acids, esters, and amides, which are compatible with the structure of cyclic lipopeptides.

Similar standards were observed in the study by Durval et al. [62], who reported signals in the range of 10 to 40 ppm, attributed to aliphatic carbons, as well as a characteristic peak between 170 and 180 ppm corresponding to ester groups. Additionally, Kumari et al. [63] identified signals compatible with iso and anteiso branching, which are structural features of branched fatty acids widely present in bacterial biosurfactants.

The structural evidence of the biosurfactant produced by *Bacillus subtilis* UCP 1533, obtained through FT-IR, ^1^H NMR, and ^13^C NMR analyses, revealed typical signals of cyclic lipopeptides, consistent with the surfactin class. In the FT-IR spectrum, characteristic absorption bands were observed at 2924 and 2853 cm^−1^ (aliphatic C–H stretching), 1743 cm^−1^ (ester carbonyl group), and 1612–1460 cm^−1^ (peptide/amide bonds), which are similar to the patterns reported for surfactin [64].

The ^1^H NMR spectrum exhibited well-defined peaks in the regions of 0.8–1.5 ppm (aliphatic chains) and 3.5–5.0 ppm (hydrogen atoms and carbons bonded to O/N), which are also described in pure surfactin compounds [65,66]. In the ^13^C NMR spectrum, signals at 14–35 ppm (aliphatic carbons) and 170–180 ppm (carbonyl groups of ester/amide) further confirmed the presence of lipidic and peptidic components characteristic of this biosurfactant group [67].

The coherence between the FT-IR absorption at 1743 cm^−1^, the 13C NMR signals at 170–180 ppm, and the 1H NMR signals at δ 0.8–1.5 ppm strongly supports the identification of the lipopeptide structure. Collectively, these spectral data provide strong evidence for the production of a biosurfactant belonging to the surfactin class.

The signals observed just above 4.0 ppm in the NMR spectrum are consistent with alpha protons (α-CH) located on the backbone of the peptide chains, which supports the presence of lipopeptide structures. Furthermore, the absence of characteristic signals in the 6.0 to 8.5 ppm region suggests that the compound does not contain aromatic residues such as phenylalanine, tyrosine, or tryptophan.

These spectral findings are consistent with those of previous reports on *B. subtilis*, particularly those described by De Faria et al. [68] and Ma [69]. Although definitive confirmation through mass spectrometry or comparison with commercial standards is still required, the spectra obtained suggest that the biosurfactant produced by *B. subtilis* UCP 1533 belongs to the surfactin group, which is widely recognized for its strong antimicrobial activity and broad biotechnological applications.

### 3.8. Evaluation of the Biosurfactant Toxicity

#### 3.8.1. Phytotoxicity

Seed germination rates and root growth of *Solanum lycopersicum* (tomato) and *Lactuca sativa* (lettuce) were used to assess the phytotoxicity of the biosurfactant produced. According to Tiquia, Tam & Hodgkiss [25], germination indices equal to or greater than 80% are indicative of the absence of phytotoxicity.

The seed germination rates of *Solanum lycopersicum* and *Lactuca sativa* showed promising results in the evaluation of the biosurfactant produced by *B. subtilis* UCP 1533, demonstrating its potential for application at controlled concentrations (Table 6). For tomato, a germination index above 80% was observed at all tested concentrations (½ CMC, 1 CMC, and 2 CMC), indicating that the biosurfactant, even at varying concentrations, was non-phytotoxic and fully compatible with the early development of this species.

For lettuce (*Lactuca sativa*), the results were also positive, with all germination values exceeding the 80% threshold, which was established as indicative of no phytotoxicity. At ½ CMC, a germination rate of 83% was observed, while at 1 and 2 CMC, the rates were 87% and 90%, respectively. This progressive increase in germination with an increase in biosurfactant concentration may indicate a stimulatory effect on *Lactuca sativa*, promoting germination at higher concentrations.

The results indicate that the biosurfactant produced by *B. subtilis* UCP 1533 does not exert toxic effects on the seeds of the evaluated species, reinforcing its viability for various applications. The absence of inhibitory effects, even at higher concentrations, suggests that the biomolecule is non-toxic under the tested conditions and highlights its potential as a plant growth-promoting agent.

Silva et al. [70] evaluated the phytotoxicity of the biosurfactant produced by *S. bombicola* ATCC 22214 on *Solanum lycopersicum* seeds, obtaining a germination rate of 87.90%. These results corroborate the non-toxicity of microbial surfactants.

#### 3.8.2. Toxicity to *Tenebrio molitor*

The larvae of *Tenebrio molitor* have become a widely adopted alternative model in toxicity studies due to their biological characteristics, low operational costs, and ease of handling, offering several advantages over traditional vertebrate models [71].

In this study, the toxicity of the biosurfactant produced by *Bacillus subtilis* UCP 1533 was evaluated, and a 100% survival rate was observed after 120 h of exposure. The same result was obtained for the negative control (PBS). No physical changes were observed in *T. molitor* larvae, such as alterations in coloration or mobility, which are indicators commonly associated with exposure to toxic agents (Figure 11).

Similar results have been reported in the literature by Lima et al. [72] and Silva et al. [46], who obtained 100% and 90% survival rates, respectively, in assays with *T. molitor* using microbial surfactants produced by the yeasts *Candida lipolytica* and *Candida glabrata*.

### 3.9. Antimicrobial Activity of Biosurfactant

Microbial resistance occurs naturally, being favored by the hereditary transfer of plasmids and the selective pressure exerted by antimicrobials, which eliminates susceptible bacterial populations and allows the proliferation of resistant strains. Moreover, spontaneous mutations contribute to the emergence of variants with a greater capacity to survive available treatments. In addition to this process, which is significantly accelerated by the inappropriate and indiscriminate use of antimicrobials in different contexts, the hospital environment, in particular, represents a critical scenario for the dissemination of resistance, since it harbors a high diversity of microorganisms and includes patients with compromised clinical conditions, often subjected to invasive procedures that make them immunologically weakened and, consequently, more susceptible to acquiring infections caused by resistant and multidrug-resistant pathogens [73].

Given this concerning scenario, the standardization of methods for evaluating bacterial susceptibility, such as the Rapid Antimicrobial Susceptibility Test (RAST) established by BrCAST-EUCAST [74], represents an essential advancement in laboratory diagnosis and therapeutic guidance in the face of clinically relevant microorganisms, especially those with recognized intrinsic or acquired resistance.

In this context, the search for new antimicrobial agents is essential to expand therapeutic options and effectively combat resistant strains. The results obtained in the antimicrobial activity assays demonstrated that the biosurfactant produced by *Bacillus subtilis* UCP 1533 exhibited antimicrobial action, highlighting its potential as a bioactive agent against the microorganisms tested.

*Escherichia coli* and *Pseudomonas aeruginosa* were sensitive to the biosurfactant, suggesting antimicrobial activity against these Gram-negative pathogens. However, the observed sensitivity is particularly relevant because of the well-known intrinsic resistance of these Gram-negative pathogens to multiple antimicrobial agents and the fact that the strains evaluated were isolated from clinical samples collected from a reference hospital specializing in the treatment of infectious diseases.

Similarly, the yeasts *Candida glabrata*, *Candida lipolytica*, *Candida bombicola*, and *Candida guilliermondii* also showed sensitivity to the compound, indicating a possible antifungal spectrum of action. These results suggest that the biosurfactant interacts with the cellular structure of these microorganisms, potentially compromising their viability at optimized concentrations or formulations. Thus, reinforcing the potential of the biosurfactant as an agent with a diversified spectrum of action, with promising applications in the control of both bacteria and fungi.

In a study conducted by Yuliani et al. [75], it was observed that the lipopeptide produced by the *Bacillus subtilis* C19 strain demonstrated selective antimicrobial activity, with inhibition observed only against the *Candida albicans* yeast, and no efficacy against Gram-positive or Gram-negative bacteria. In contrast, Das et al. [76] reported the antimicrobial activity of a glycolipid produced by *Bacillus* sp., with proven action against *Staphylococcus aureus* and *Escherichia coli*. In an even broader scope, the lipopeptide produced by *Pseudomonas* sp. OXDC12 was effective against a wide range of Gram-positive and Gram-negative bacteria, including *Staphylococcus aureus* MTCC96, *Salmonella typhimurium* NCTC 74, *Klebsiella pneumoniae*, and *Escherichia coli* MTCC1687 [77]. Similarly, the glycopolypeptide produced by *Bacillus pumilus* SG also showed significant antimicrobial activity against different bacterial species, such as *Acinetobacter* spp., *Escherichia coli*, *Bacillus cereus*, *Pseudomonas aeruginosa*, and *Streptococcus pneumoniae* [78]. The lipopeptide produced by *Bacillus alveayuensis* exhibited antimicrobial activity against *Desulfovibrio marinus* BRS1, a microorganism known for its resistance to marine environments and its role in microbiologically influenced corrosion [79].

It is important to highlight that this study did not include antibiotic standards as a positive control. Although this limitation is justified by the preliminary and exploratory nature of the study, the absence of comparative standards restricts a more precise assessment of the relative efficacy of the biosurfactant. Future studies should incorporate standardized antibiotics to allow for quantitative benchmarking of antimicrobial potency and better contextualization of the biotechnological potential of the biosurfactants.

Thus, the results of the present study, although not showing clear inhibition halos, are consistent with previous findings that indicate variability in the efficacy of biosurfactants, depending on their composition and the susceptibility of the target microorganisms. These observations reinforce the need for further investigation of the physicochemical characterization of the biosurfactant and exploration of other methodologies, such as minimum inhibitory concentration (MIC) tests and combinatory approaches with other antimicrobials, to better understand its potential and expand its applications.

## 4. Conclusions

This study demonstrated that the lipopeptide biosurfactant produced by *Bacillus subtilis* UCP 1533, isolated from soil in the Brazilian semi-arid region, possesses physicochemical and functional properties that qualify it as a promising and sustainable alternative with high biotechnological potential. The formulation using soybean oil and corn steep liquor not only demonstrated the feasibility of employing alternative and low-cost carbon and nitrogen sources, but also resulted in efficient biosurfactant production. This more economical and environmentally sustainable approach led to high productivity, low critical micelle concentration, and a significant reduction in surface tension.

Moreover, the biosurfactant exhibited stability across a wide range of temperatures, pH values, and salinities, which is highly desirable and decisive for its application in various fields. It also exhibited high emulsifying activity against hydrocarbons. Toxicity assays revealed no adverse effects in invertebrate and plant models, further reinforcing their safety for biotechnological application.

Additionally, the observed antimicrobial activity against multidrug-resistant pathogens, such as *Escherichia coli* and *Pseudomonas aeruginosa*, as well as yeasts of the *Candida* genus, indicates its potential as an adjuvant agent in microbial control strategies.

The results obtained indicate that the biosurfactant produced by *B. subtilis* UCP 1533 is a sustainable and effective alternative to synthetic surfactants, with promising applications in industrial processes and antimicrobial formulations. However, further in-depth studies are needed to explore additional analytical and microbiological methodologies, including minimum inhibitory concentration (MIC) and minimum bactericidal concentration (MBC) assays, as well as synergy tests with conventional antimicrobials. These approaches will contribute to a better understanding of the mechanism of action and may help broaden the spectrum of activity of the biosurfactant.

## Figures and Tables

**Figure 1 microorganisms-13-01548-f001:**
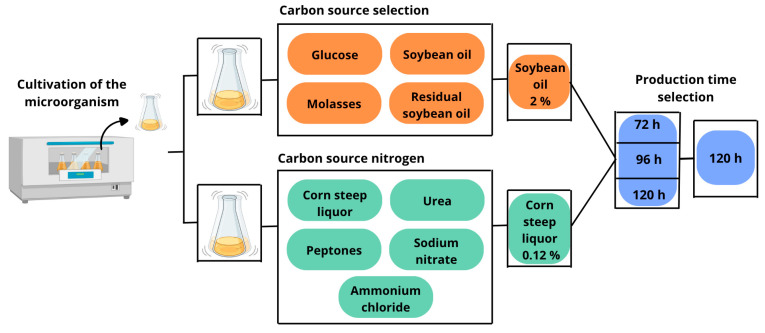
Flowchart of cultivation and optimization of biosurfactant production by *Bacillus subtilis.* The microorganism was grown in a mineral medium containing inorganic salts, Tris buffer, and yeast extract. Fermentation was carried out in 250 mL Erlenmeyer flasks with 100 mL of medium at 35 °C and 200 rpm for 120 h. Different carbon sources (glucose, molasses, soybean oil, and used frying oil) and nitrogen sources (corn steep liquor, urea, ammonium chloride, sodium nitrate, and peptones) were evaluated for medium optimization.

**Figure 2 microorganisms-13-01548-f002:**
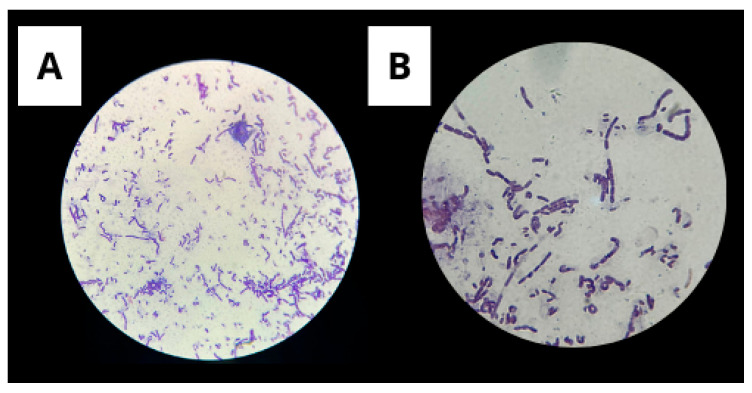
Gram staining showing Gram-positive bacilli. (**A**) Image captured with a 40× objective; (**B**) Image captured with a 100× objective (oil immersion).

**Figure 3 microorganisms-13-01548-f003:**
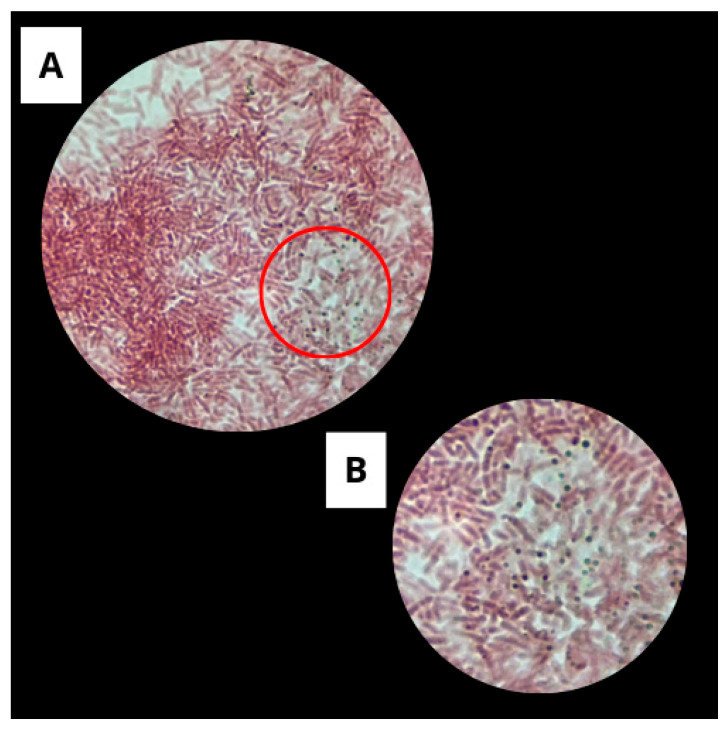
Spore staining using the Wirtz-Conklin method. (**A**) Overview under 100× objective, showing pink-stained cells. (**B**) Magnification of the highlighted region, revealing oval spores stained light green inside the cells.

**Figure 4 microorganisms-13-01548-f004:**
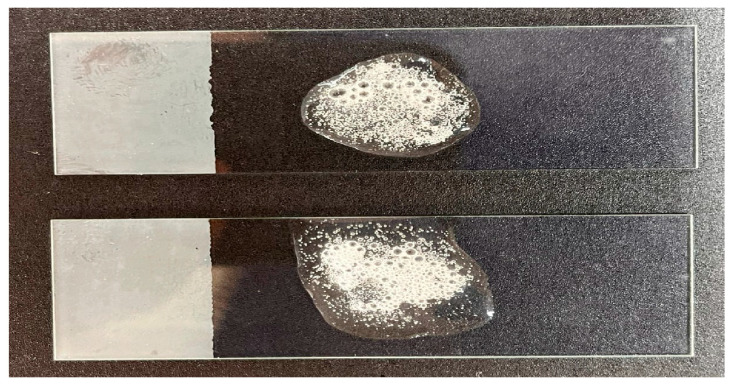
The Catalase test was performed in duplicate. Bubble formation was observed on both slides after the addition of hydrogen peroxide (H_2_O_2_) solution, indicating the presence of catalase.

**Figure 5 microorganisms-13-01548-f005:**
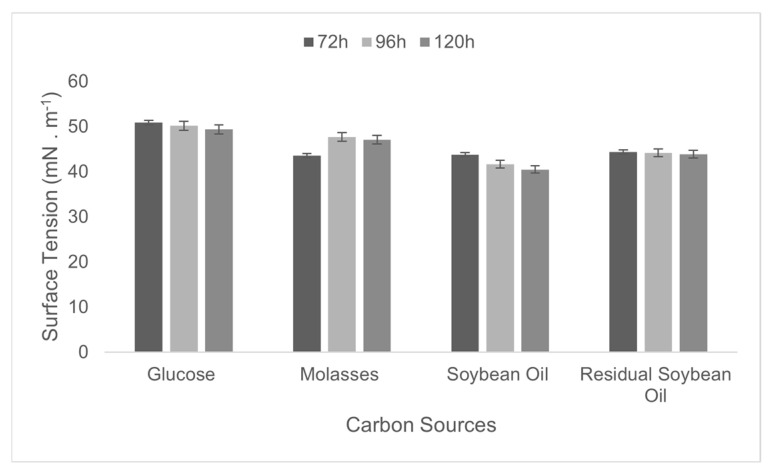
Variation in surface tension (mN·m^−1^) over time (72, 96, and 120 h) in culture media containing different carbon sources.

**Figure 6 microorganisms-13-01548-f006:**
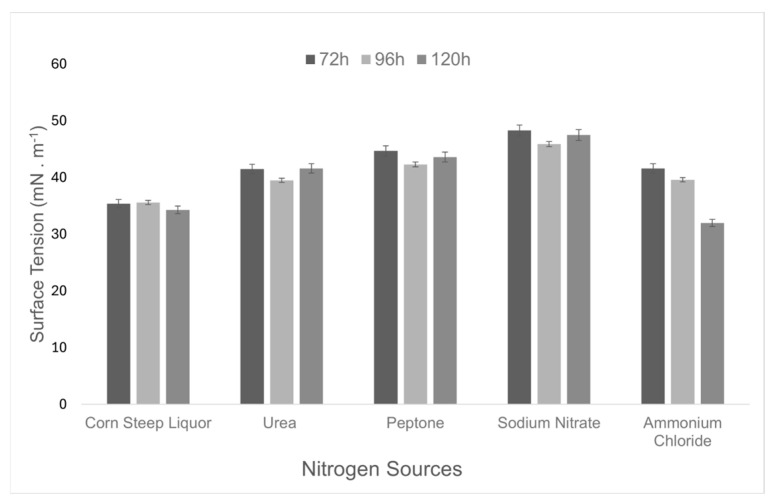
Variation in surface tension (mN·m^−1^) over time (72, 96, and 120 h) in culture media containing different nitrogen sources.

**Figure 7 microorganisms-13-01548-f007:**
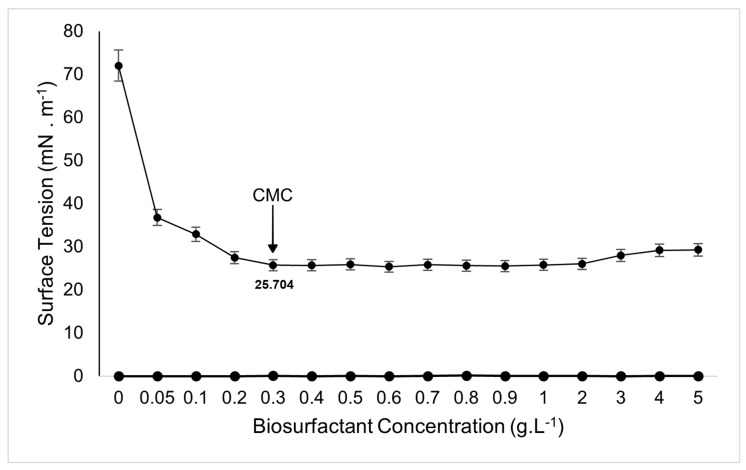
Variation of surface tension as a function of the concentration of biosurfactant produced by *Bacillus subtilis* UCP 1533. The critical micelle concentration (CMC) was determined as the point beyond which increases in concentration no longer resulted in significant reductions in surface tension, indicating the minimum concentration required for micelle formation.

**Figure 8 microorganisms-13-01548-f008:**
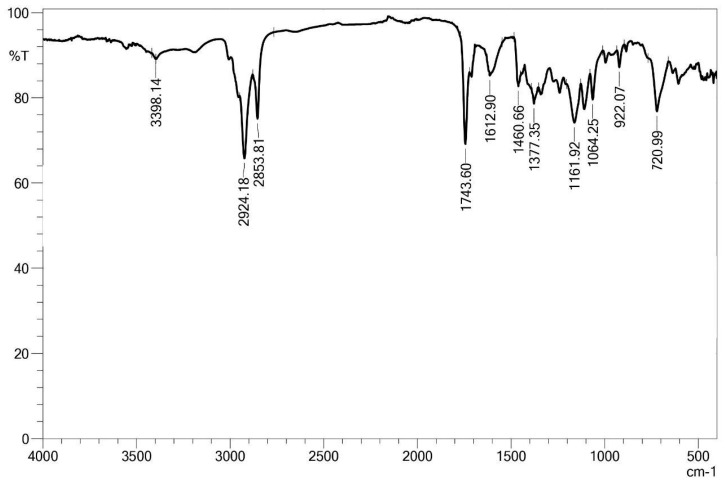
FT-IR spectrum of isolated biosurfactant. The *x*-axis represents the wavenumber (cm^−1^), and the *y*-axis represents the transmittance (%).

**Figure 9 microorganisms-13-01548-f009:**
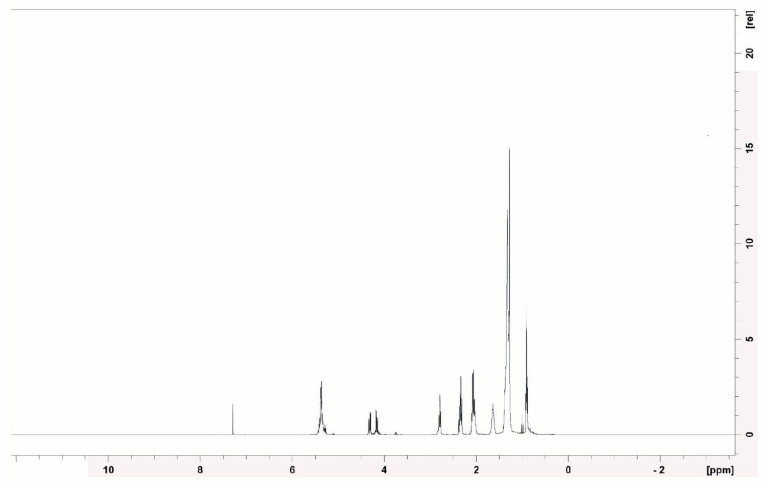
^1^H NMR spectrum of the isolated biosurfactant.

**Figure 10 microorganisms-13-01548-f010:**
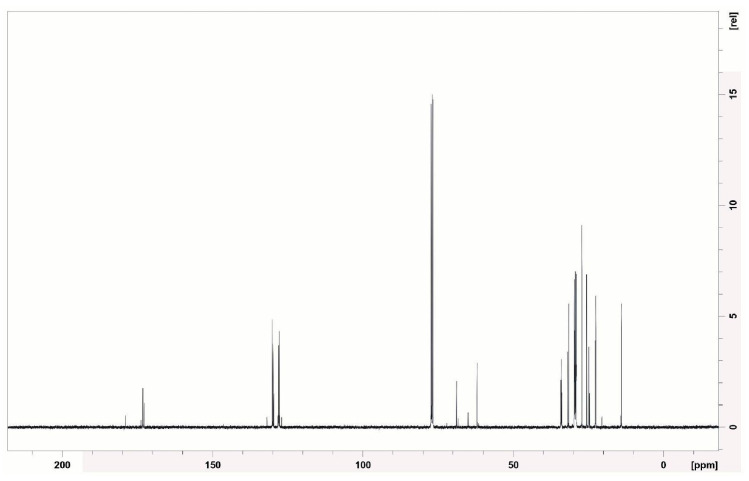
^13^C NMR spectrum of the isolated biosurfactant.

**Figure 11 microorganisms-13-01548-f011:**
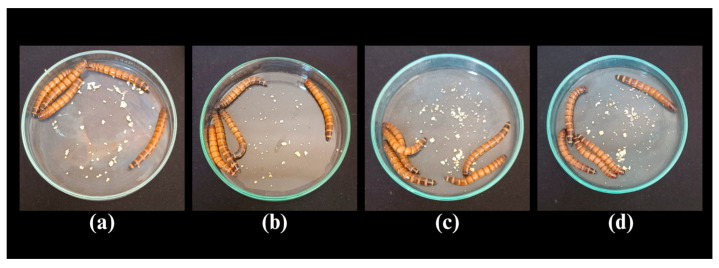
Images of *Tenebrio molitor* larvae after exposure to phosphate-buffered saline (PBS) and different biosurfactant concentrations during the toxicity assay. (**a**) 1/2 CMC; (**b**) 1 CMC; (**c**) 2 CMC; (**d**) Negative control (PBS).

**Table 1 microorganisms-13-01548-t001:** Surface tension (mN·m^−1^) of the growing mediums after 120 h, using different carbon and nitrogen sources.

Evaluated Sources	Substrate	Concentration	Surface Tension (mN·m^−1^) of Best Growing Time (120 h)
	Glucose		49.4 ± 0.004
	Molasses	2%	47.1 ± 0.002
Carbon	Soybean oil		40.5 ± 0.012
	Residual soybean oil		43.9 ± 0.009
	Corn steep liquor		34.3 ± 0.008
	Urea		41.6 ± 0.003
Nitrogen	Peptone	0.12%	43.6 ± 0.002
	Sodium nitrate		47.5 ± 0.009
	Ammonium chloride		32.0 ± 0.004

**Table 2 microorganisms-13-01548-t002:** Surface tension stability of the biosurfactant produced by *B. subtilis* UCP 1533.

Temperature (°C)	Surface Tension mN·m^−1^	pH	Surface Tension mN·m^−1^	NaCl (%)	Surface Tension mN·m^−1^
5	31.9 ± 0.09	2	30.7 ± 0.08	2	27.9 ± 0.06
50	26.0 ± 0.05	4	27.7 ± 0.05	4	27.4 ± 0.07
70	27.7 ± 0.08	6	30.8 ± 0.02	6	27.4 ± 0.04
100	26.2 ± 0.05	8	26.8 ± 0.05	8	27.8 ± 0.08
		10	29.5 ± 0.07	10	27.4 ± 0.08
		12	28.4 ± 0.03	12	27.7 ± 0.04

**Table 3 microorganisms-13-01548-t003:** Stability of emulsification of biosurfactant produced by *B. subtilis* UCP 1533 at different pH values.

pH	Motor Oil (%)	Soybean Oil (%)	Sunflower Oil (%)
2	100 ± 0.01	50 ± 0.03	35 ± 0.01
4	100 ± 0.02	55 ± 0.02	40 ± 0.02
6	100 ± 0.02	55 ± 0.02	40 ± 0.02
8	100 ± 0.01	55 ± 0.01	40 ± 0.02
10	100 ± 0.01	55 ± 0.02	40 ± 0.01
12	100 ± 0.02	55 ± 0.01	40 ± 0.02

**Table 4 microorganisms-13-01548-t004:** Emulsification stability of biosurfactants produced by *B. subtilis* UCP 1533 at different temperatures.

Temperature (°C)	Motor Oil (%)	Soybean Oil (%)	Sunflower Oil (%)
5	100 ± 0.02	60 ± 0.03	35 ± 0.04
50	100 ± 0.01	55 ± 0.02	50 ± 0.03
70	100 ± 0.01	60 ± 0.01	55 ± 0.03
100	100 ± 0.02	60 ± 0.02	50 ± 0.02

**Table 5 microorganisms-13-01548-t005:** Emulsification stability of biosurfactants produced by *B. subtilis* UCP 1533 at different NaCl concentrations.

NaCl (%)	Motor Oil (%)	Soybean Oil (%)	Sunflower Oil (%)
2	100 ± 0.03	40 ± 0.04	30 ± 0.01
4	100 ± 0.01	40 ± 0.05	30 ± 0.03
6	100 ± 0.02	40 ± 0.04	35 ± 0.02
8	100 ± 0.02	40 ± 0.04	30 ± 0.01
10	100 ± 0.01	45 ± 0.02	30 ± 0.04
12	100 ± 0.02	45 ± 0.04	30 ± 0.03

**Table 6 microorganisms-13-01548-t006:** Germination indices of *Solanum lycopersicum* and *Lactuca sativa* at different concentrations of biosurfactant produced by *Bacillus subtilis* UCP 1533.

Concentration of Biosurfactant	Tomato (*Solanum lycopersicum*) (%)	Lettuce (*Lactuca sativa*) (%)
Distilled water	90	85
½ CMC	80	83
1 CMC	90	87
2 CMC	80	90

## Data Availability

Data are contained within the article.

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
