# Peer review of "Biosurfactant Produced by Bacillus subtilis UCP 1533 Isolated from the Brazilian Semiarid Region: Characterization and Antimicrobial Potential"

_microorganisms, 2025, doi:10.3390/microorganisms13071548_

Round 1

Reviewer 1 Report

Comments and Suggestions for Authors

Please find attached reviewers comments

Author Response

Dear reviewer,

The  authors would like to thank you for your constructive comments on our manuscript and we have covered all of your comments the best of our knowledge.

Best regards,

Reviewer 2 Report

Comments and Suggestions for Authors

This manuscript reports the isolation, production optimization, physicochemical characterization, and antimicrobial evaluation of a biosurfactant produced by Bacillus subtilis UCP 1533 from the Caatinga soil. This topic is relevant and timely, given the global push toward sustainable and biologically derived antimicrobial agents. The authors combined classical microbiological methods with spectroscopic analysis and bioassays to yield a rich dataset.

The experimental design was generally sound and the presentation was thorough. However, the manuscript requires major revisions to address issues related to overinterpretation of results, methodological clarification (especially controls), language polishing, and presentation of key findings. Some data should be visualized more effectively (e.g., surface tension trends), and comparisons with standard antimicrobial agents should be strengthened.

  • Line 2: Consider shortening the title by removing the phrase “isolated from the Caatinga soil…” and placing it in the abstract or keywords.
  • Lines 14–28: The abstract is well structured but overly packed with numerical data. This suggests reducing numerical overload (e.g., specific temperatures and % salinity) and highlighting only the key outcomes.
  • Lines 30–36: Add a sentence on potential applications (e.g., agriculture and pharmaceuticals) to strengthen this impact.
  • Lines 38–60: The introduction appropriately contextualizes AMR, but lines 48–60 repeat the concepts. Streamlines to reduce redundancy in antimicrobial resistance drivers.
  • Line 61: Clarify “economic barriers”.
  • Lines 153–156: The catalase test procedure is clear but considers referencing CLSI or ISO microbial identification guidelines.
  • Lines 165–185: The explanation of cultivation condition optimization is commendably detailed, but verbose. Suggest summarizing steps into fewer lines or workflow figure.
  • Lines 230–231: Biosurfactant extraction protocol is clear and replicable. Minor stylistic edit: “subjected to evaporation on a heated plate” to “evaporated using a heated plate.”
  • Line 245: The ionic charge assay is novel and appropriate. Please cite the relevant primary literature, rather than adapting solely from Luna et al. [21].
  • Lines 252–267: Toxicity and phytotoxicity assays are relevant and clearly described. You may mention the ethical handling of T. molitor larvae in accordance with the institutional policy, if applicable.
  • Lines 268–282: The disk diffusion method is appropriate, but a comparison with reference antibiotics (e.g., positive controls) should be included in the context.
  • Lines 290–307: The MALDI-TOF identification process is justified. A score of 1.87 indicates genus-level ID; you should mention if sequencing (e.g., 16S rRNA) was considered for confirmation.
  • Lines 320–341: Wirtz-Conklin staining results are consistent. Add a comment on sporulation frequency or abundance, if quantified.
  • Lines 358: Rephrase “provide a robust set of evidence” → “provide supportive evidence.”
  • Lines 374–379: Repetition of "surface tension" as a key parameter is excessive. Summarize this with one definitive sentence.
  • Line 495: Biosurfactant stability results are informative. You claim “moderate” and “excellent” stability—please define criteria (e.g., % variation threshold).
  • Lines 621–630: Clear identification of the lipopeptide nature. Suggests including NMR peak range examples to support the FTIR interpretation.
  • Line 647: The statement regarding the biostimulant effect is speculative. Consider softening to “may indicate a stimulatory effect…”
  • Line 654: Excellent section. Rephrase “demonstrates the safety of the biomolecule” to “suggests the biomolecule is non-toxic under tested conditions.”
  • Lines 682–698: Disk diffusion outcomes are presented well. Consider comparing inhibition zone sizes to standard antimicrobials for better interpretability.
  • Lines 748–750: The statement calling for MIC testing is both appropriate and realistic. Suggest adding: “Future studies should include MIC, MBC, and synergy assays.”
  • Table 2: Well-structured. Consider plotting the surface tension across the pH/temperature as graphs for visual clarity.
  • Table 6: Indicates phytotoxicity. Consider the addition of a control row (distilled water only) for comparison.
  • Lines 770 onward: References are appropriate and current. Be sure to format consistently (e.g., remove double hyphens in page ranges like “810–-830”).

Author Response

Dear Reviewer,

The authors would like to thank you for your constructive comments on our manuscript and we have covered all of your comments  to the best of our knowledge.

Best regards,

Reviewer 3 Report

Comments and Suggestions for Authors

This manuscript presents the production, characterization, and antimicrobial activity of a new biosurfactant. The writing and explanations are clear and the characterization methods are rigorous. I recommend publication after the authors address the following questions/comments related to the characterization of the biosurfactant.

In Figures 4 and 5, were soybean oil and corn steep liquor chosen because the surfactant produced with these carbon and nitrogen sources gave the lowest surface tensions? Many of the surface tensions in Table 1 are similar. Could the authors clarify the criteria used?

Section 3.3 presents surfactant yield and production. The biosurfactant is also identified as a lipopeptide. Do the results, though, show that the biosurfactant is a single pure compound? Could instead a mixture of lipopeptides with similar structures been produced? For example, mixture components could have different amino acids in the headgroup or different degrees of saturation in the hydrocarbon chain. Did the authors do TLC or another similar experiment to show that the biosurfactant was a pure substance?

In section 3.4 a CMC of 0.3 g.L-1 is reported. This value was also reported to be superior to another biosurfactant from the literature which had a higher CMC. Could the authors clarify why a lower CMC is preferred? In some application like those involving the absorption of surfactant monomers, higher CMC’s may give better performance. Also in Figure 6, the CMC is identified as the point where the surface tension begins to increase. Typically, CMC’s are measured by curve fitting data at high and low concentrations and identifying the intersection of the curves as the CMC. Using curve fitting to determine the CMC would likely be less subjective and lead to a more accurate value.

In Tables 3-5 values are reported to one or two significant figures, but errors are in the +/- 0.01-0.04 range. Does this imply in Table 3 for example that the percentages are 100.00 +/- 0.01 or is the 0.01 reporting a 1% error in the measurement?

In the paragraph beginning on line 591, -CH2-CO was referred to as a peptide bond. A peptide bond though typically refers to the amide linkage between two amino acids. Could the authors clarify this point?

The NMR spectrum in Figure 8 shows strong evidence of a lipopeptide. Can the peaks just above 4.0 ppm be identified as the alpha-protons in the peptide backbone? Also it may be important to note that the NMR spectrum contains no aromatic resonances, so the lipopeptide must not contain amino acids like phenylalanine or tyrosine with aromatic side chains.

Finally, the biosurfactant surfactin is mentioned in the Introduction. This biosurfactant contains a cyclic peptide headgroup and an aliphatic chain. Is there reason to suspect that given the microorganisms used or based on the characterization experiments that the biosurfactant produced in this study had a similar structure?

This manuscript presents the production, characterization, and antimicrobial activity of a new biosurfactant. The writing and explanations are clear and the characterization methods are rigorous. I recommend publication after the authors address the following questions/comments related to the characterization of the biosurfactant.

In Figures 4 and 5, were soybean oil and corn steep liquor chosen because the surfactant produced with these carbon and nitrogen sources gave the lowest surface tensions? Many of the surface tensions in Table 1 are similar. Could the authors clarify the criteria used?

Section 3.3 presents surfactant yield and production. The biosurfactant is also identified as a lipopeptide. Do the results, though, show that the biosurfactant is a single pure compound? Could instead a mixture of lipopeptides with similar structures been produced? For example, mixture components could have different amino acids in the headgroup or different degrees of saturation in the hydrocarbon chain. Did the authors do TLC or another similar experiment to show that the biosurfactant was a pure substance?

In section 3.4 a CMC of 0.3 g.L-1 is reported. This value was also reported to be superior to another biosurfactant from the literature which had a higher CMC. Could the authors clarify why a lower CMC is preferred? In some application like those involving the absorption of surfactant monomers, higher CMC’s may give better performance. Also in Figure 6, the CMC is identified as the point where the surface tension begins to increase. Typically, CMC’s are measured by curve fitting data at high and low concentrations and identifying the intersection of the curves as the CMC. Using curve fitting to determine the CMC would likely be less subjective and lead to a more accurate value.

In Tables 3-5 values are reported to one or two significant figures, but errors are in the +/- 0.01-0.04 range. Does this imply in Table 3 for example that the percentages are 100.00 +/- 0.01 or is the 0.01 reporting a 1% error in the measurement?

In the paragraph beginning on line 591, -CH2-CO was referred to as a peptide bond. A peptide bond though typically refers to the amide linkage between two amino acids. Could the authors clarify this point?

The NMR spectrum in Figure 8 shows strong evidence of a lipopeptide. Can the peaks just above 4.0 ppm be identified as the alpha-protons in the peptide backbone? Also it may be important to note that the NMR spectrum contains no aromatic resonances, so the lipopeptide must not contain amino acids like phenylalanine or tyrosine with aromatic side chains.

Finally, the biosurfactant surfactin is mentioned in the Introduction. This biosurfactant contains a cyclic peptide headgroup and an aliphatic chain. Is there reason to suspect that given the microorganisms used or based on the characterization experiments that the biosurfactant produced in this study had a similar structure?

Author Response

Dear Reviewer,

The authors would like to thank you for your constructive comments on our manuscript and we have covered all of your comments  to the best of our knowledge.

Reviewer 4 Report

Comments and Suggestions for Authors

The authors investigated a biosurfactant produced by Bacillus subtilis UCP 1533, isolated from the Brazilian semiarid soil, aiming to characterize its properties and evaluate its antimicrobial potential as a safe and sustainable alternative against resistant pathogens. The bacterial strain was confirmed as Bacillus subtilis through MALDI-TOF MS, Gram staining (Gram-positive rods), spore staining, and catalase tests. They examined the characteristics of biosurfactant in terms of optimal production conditions, emulsifying activity, stability across wide ranges of pH (2-12), salinity (2-12% NaCl), and temperature (5-100°C). Spectroscopic analyses (FT-IR and NMR) confirmed its lipopeptide nature and anionic character.
Toxicity assessments revealed the biosurfactant's low toxicity, with 100% survival observed in Tenebrio molitor larvae and high germination rates (over 80%) for tomato and lettuce seeds, suggesting it is non-phytotoxic and potentially a biostimulant. Furthermore, the biosurfactant displayed antimicrobial activity against clinical strains of multidrug-resistant Escherichia coli and Pseudomonas aeruginosa, as well as various Candida species (C. glabrata, C. lipolytica, C. bombicola, C. guilliermondii). This indicates its significant potential in combating antimicrobial resistance (AMR).
In conclusion, the biosurfactant produced by Bacillus subtilis UCP 1533 is a safe, effective, and sustainable alternative to synthetic surfactants, offering promising applications in diverse industrial processes and in the ongoing fight against antimicrobial resistance.

The paper has been well-written and well-structured. There have been many reports on the biosurfactant produced by Bacillus subtilis. Therefore, whole part of the submitted manuscript seemed to be mundane. I think the submitted manuscript would be improved if the authors try to highlight the importance of this study. For example, adding several descriptions on which point is novel things or superior to other biosurfactants produced by B. subtilis.

There were several minor points to be corrected or revised in the submitted manuscript. Please consider the following things.

Line 107 : Does “0.85%” mean 0.85%(w/v)? If so, please add “(w/v)” after %. Please check again the definition of percent (%).

Lines 154 and 344 : Please add the word "solution" after hydrogen peroxide. Hydrogen peroxide (H2O2) is the name of a chemical compound. In this paper, concentrated hydrogen peroxide is thought to be dissolved in water to achieve a final concentration of 3% (w/v or v/v), rather than using the pure chemical compound.

Lines 166-168, 179, 182, 192, 193, 400, 401, 531, 547 : Do those “%” mean (w/v)? If so, please add “(w/v)” after %. Please check again the definition of percent (%).

Line 196 : Please describe the maker name and model of the orbital shaker.

Lines 219-220 : Do those “%” mean (w/v)? Please check again the definition of percent (%).

Line 227 : Please express centrifugation conditions as RCF (Relative Centrifugal Force) instead of 4,500 rpm; i.e., ......... × g (where g is in italics).

Figures 4, 5 and 6 : Please check the description of the unit on the Y-axis; the position of the dot (•) between mN and m-1. In addition, "-1" after m should be written as a superscript.

Figure 7 : The x-axis is "Wavenumber (cm-1)" and "-1" is written as a superscript. The y-axis is "Transmittance (%)."

Line 388 : Plural of medium is “media.”

Line 560 : "3.7. Structural characterization of the biosurfactant produced by Bacillus subtilis UCP 1533"
The authors conducted structural characterization of the biosurfactant using FT-IR and NMR. Can the molecular structure of the biosurfactant produced by B. subtilis UCP 1533 be predicted and categorized? Surfactin, iturin, and fengycin have been reported as biosurfactants produced by B. subtilis. It would be beneficial to describe in the discussion which biosurfactant group the compound produced by B. subtilis UCP 1533 belongs to.

Lines 663-667 : The authors have described as follows: ”In this study, the toxicity of the biosurfactant produced by Bacillus subtilis UCP 1533 was evaluated, and a 100 % survival rate was observed after 120 h of exposure. The same result was obtained with the negative control (PBS). No physical changes were observed in the T. molitor larvae, such as alterations in coloration or mobility, which are indicators commonly associated with exposure to toxic agents.” 

I think it would be helpful to show visual results such as pictures of the biosurfactant-treated and negative control (PBS) groups, given that no physical changes were observed in the T. molitor larvae. According to the description in the text, no physical changes between these groups should be observed.

Author Response

(The authors gave the same response as above.)

Round 2

Reviewer 2 Report

Comments and Suggestions for Authors

Dear Authors,

Thank you for submitting the revised version of your manuscript titled “Biosurfactant produced by Bacillus subtilis UCP 1533 isolated of the Brazilian semiarid region: characterization and antimicrobial potential.” I appreciate the detailed and thoughtful responses provided in your rebuttal letter.

You have successfully addressed the key concerns raised in the previous round of review. Specifically:

  • The abstract has been refined for clarity and has reduced redundancy.
  • The title has been shortened and streamlined without compromising the scientific accuracy.
  • Methodological explanations, such as those related to catalase testing, ionic charge determination, and larval handling, were appropriately clarified.
  • The inclusion of a distilled water control in the phytotoxicity assessment and the addition of peak regions in the NMR section improved the scientific depth of the study.
  • The language in the Discussion and Conclusion sections has been revised to maintain caution in interpreting safety and biostimulant effects.

Most notably, the definition of biosurfactant stability categories and the incorporation of a visual schematic (flowchart) enhanced the accessibility of the manuscript.

Only one point—the absence of antibiotic standards in the antimicrobial assay—was not addressed experimentally in this study. Although the justification provided (i.e., preliminary scope) is understandable, this limitation should be clearly acknowledged in the discussion or limitations section of the manuscript for transparency.

The manuscript now presents a coherent and scientifically sound preliminary report on a promising biosurfactant-producing strain with ecological relevance and potential for biotechnological applications.

Recommendation: Accept after minor language corrections (optional, no further experimental changes required).

With best regards,

Author Response

(The authors gave the same response as above.)
